# Application of Health Belief Model for the assessment of COVID-19 preventive behavior and its determinants among students: A structural equation modeling analysis

**Kegnie Shitu[1]\*, Asmamaw Adugna[1], Ayenew Kassie[1], Simegnew Handebo[2]**

1 Department of Health Education and Behavioral Sciences, Institute of Public Health, College of Medicine and Health Sciences, University of Gondar, Gondar, Ethiopia, 2 School of Public Health, St. Paul's Hospital Millennium Medical College, Addis Ababa, Ethiopia

\* kegnsh@gmail.com, kegnie.shitu@uog.edu.et

**Data Availability Statement:** The datasets generated and/or analyzed in the current study are

## Abstract

### Background

COVID-19 is a new pandemic that poses a threat to people globally. In Ethiopia, where classrooms are limited, students are at higher risk for COVID-19 unless they take consistent preventative actions. However, there is a lack of evidence in the study area regarding student compliance with COVID-19 preventive behavior (CPB) and its predictors.

### Objective

This study aimed to assess CPB and its predictors among students based on the perspective of the Health Belief Model (HBM).

### Method and materials

A school-based cross-sectional survey was conducted from November to December 2020 to evaluate the determinants of CPB among high school students using a self-administered structured questionnaire. 370 participants were selected using stratified simple random sampling. Descriptive statistics were used to summarize data, and partial least squares structural equation modeling (PLS-SEM) analyses to evaluate the measurement and structural models proposed by the HBM and to identify associations between HBM variables. A T-value of > 1.96 with 95% CI and a P-value of < 0.05 were used to declare the statistical significance of path coefficients.

### Result

A total of 370 students participated with a response rate of 92%. The median (interquartile range) age of the participants (51.9% females) was 18 (2) years. Only 97 (26.2%), 121 (32.7%), and 108 (29.2%) of the students had good practice in keeping physical distance, frequent hand washing, and facemask use respectively. The HBM explained 43% of the variance in CPB. Perceived barrier ($\beta = -0.15$, $p < 0.001$) and self-efficacy ($\beta = 0.51$, $p < 0.001$)

available in DRYAD at the DOI https://doi.org/10.5061/dryad.dbrv15f2x.

**Funding:** The author(s) received no specific funding for this work.

**Competing interests:** The authors have declared that no competing interests exist.

**Abbreviations:** AVE, Average Variance Extracted; CR, Composite Reliability; CA, Cues to action; CBM, COVID-19 preventive behaviors; HTMT, Hetrotraite-Monotraite; PBA, Perceived barrier; PBE, Perceived benefit; PSE, Perceived severity; SEF, Self efficacy.

were significant predictors of student compliance to CPB. Moreover, the measurement model demonstrated that the instrument had acceptable reliability and validity.

## Conclusion and recommendations

COVID-19 prevention practice is quite low among students. HBM demonstrated adequate predictive utility in predicting CPBs among students, where perceived barriers and self-efficacy emerged as significant predictors of CPBs. According to the findings of this study, theory-based behavioral change interventions are urgently required for students to improve their prevention practice. Furthermore, these interventions will be effective if they are designed to remove barriers to CPBs and improve students' self-efficacy in taking preventive measures.

## Introduction

The global outbreak of the COVID-19 pandemic has spread worldwide, affecting almost all countries and territories [1]. The pandemic is highly contagious that has been spread to different countries within a few months of its occurrence and it was declared as a pandemic by the World Health Organization (WHO) making it the concern of the global population [1, 2].

In addition to causing morbidities and mortalities, the pandemic affects all aspects of human life [3, 4]. Education is one of the most affected systems worldwide by the pandemic [5]. Consequently, schools were locked down for months by which students were obliged to stay at home with no education. Its effect on education is disproportionately higher in developing countries like Ethiopia due to the poor technological infrastructures that could support online education [6, 7].

The federal government of Ethiopia declared school reopening after six months of lock down so that students have started face to face leaning [8]. Students learn in a crowded environment because of the limited number of classrooms in schools, which can increase the risk of COVID-19 transmission unless appropriate preventive measures are taken [9]. Given this, the involvement of students in COVID-19 preventive activities is crucial to contain the spread of the pandemic among themselves and their community at large.

Even though there is no a proven drug to treat COVID-19 [10], there are various nonpharmaceutical interventions such as hand-washing, wearing a facemask, and physical distancing have established their relevance in preventing the spread of the infection [1]. On the contrary, schools are institutions where students are gathered together to learn, which makes physical distancing difficult and in turn imposes an increased risk of COVID-19 infection unless adequate precaution measures are taken by the schools and the students too [11].

Studies from various parts of the world have shown that student's engagement in COVID-19 preventive behaviors is highly variable across countries. For example, 74.5% of students wear a facemask and 85% of students washed their hands frequently in Bangladesh [12], 80.0% of students adopted social isolation strategies, regular hand washing, and enhanced personal hygiene measures in Jordan [13] and more than 94% of students were following the recommended preventive health behaviors in Iran [14]. On the other hand, few studies among Ethiopian university students have shown unsatisfactory compliance behavior of students towards COVID 19 preventive measures, where only 56% of students keep their physical distance, 74% wash their hands frequently [15, 16].

COVID-19 is not only a health threat, but it is also a threat to the socio-economic aspect of human life [17].

The extent to which a disaster like the COVID-19 pandemic could impact people's lives depends on how they perceive the disaster [18, 19]. For example, a study done by Mahmoud AB and his colleagues claimed that individuals' job insecurity was highly dependent on their perceptions of the pandemic in which employees with an intense perception of COVID-19 were more likely to be exposed to job insecurity [19].

Risk reduction measures such as social distancing, hand washing, and wearing a facemask can rarely be enforced entirely by coercion unless students must understand what is expected of them and feel strongly about the importance of compliance. In this regard, Health Belief Model (HBM) is the most appropriate fit behavioral framework to understand why students are /aren't participating in COVID-19 preventive measures [20].

The HBM was developed in the 1950s for the purpose to explain why people do/don't take a certain preventive measure if they face the risk of being ill [20–22]. According to this model, students are most likely to take COVID-19 preventative measures if they perceive the threat of contracting the infection is to be serious, feel they are personally susceptible to the infection, have the confidence of executing the recommended preventive actions, and perceive that there are fewer costs than benefits to engaging in preventive measures [23].

HBM has shown an adequate utility in the prediction of the various spectrum of health behaviors [21, 22, 24]. For instance, studies from India [25], Iran [26], Iraq [27], and Ethiopia [28, 29] have shown the predictive utility of the model in the prediction of COVID-19 preventive health behaviors among the various population.

Theory, research, and practice are interrelated concepts that are essential to understanding health behavior and health behavior change. The best theory is informed by practice and the best practice should be grounded in theory [30]. The HBM is one of the various health behavior models that has been demonstrated its importance in guiding behavior change intervention in various settings [31, 32]. This model also could help us to understand why students do/don't take COVID-19 preventive measures by taking their perception of the pandemic and the recommended measures into account.

Moreover, identifying important cognitive factors that drive the students' compliance to the preventive measures of COVID-19 would have greater importance to design appropriate health communication programs that influence students to take the recommended preventive measures of COVID-19. Furthermore, students are at higher risk of getting COVID-19 infection due to the large class size, a common phenomenon in resource-limited countries like Ethiopia. In this case, following the recommended measures for COVID-19 infection is of great importance. However, students' COVID-19 prevention practice and associated factors have remained unstudied in the academia. Therefore, this study aimed to assess COVID-19 preventive health behavior (CPB) and associated factors among secondary school students in Gondar City, North West Ethiopia based on the Health Belief Model [S1 Fig]. Furthermore, the authors believe that the study generated a shred of preliminary evidence that could have paramount importance to design contextual behavior change interventions among students in the study area to improve student's COVID-19 prevention practice.

## Methods

### Study participants

An institution based crossectional study was conducted from November to December 2020 among secondary school students (Grade 9th to 12th) attending their class during the academic year of 2020/2021in Gondar city. According to the Ethiopian education system, secondary

school students refer to students who attend their class at high school (9th to 10th) and preparatory schools (11th to 12th). All secondary school students who were attending their class at Gondar city were included in the present study. Students who are out of school during the data collection period after home checkups (A family call to check whether students unavailable at school during data collection were available at home and able to fill a questionnaire) were excluded from the study. In addition to this, students who were transferred out and/or transferred in from schools out of Gondar in the 2020/2021 academic year were also excluded.

## Sample size determination and procedure

The sample size was determined for another study which is submitted to BMC Journal of Psychiatry that aimed to assess COVID-19 associated anxiety among students. Moreover, the sample size was also above the minimum required sample size for PLS-SEM analysis based on the ten times rule of thumb [33]. Our final model consisted of a total of 32 (5 in the inner model and 27 in the outer model) paths indicated that at least 320 (10 times 32) observations are required for PLS-SEM estimations. Fortunately, our pre-calculated sample was 370 which was adequate to proceed with PLS-SEM. Furthermore, we conducted a post hoc power analysis to verify that the sample was adequate to detect the required estimates precisely based on the following assumption: the amount of explained variance in the endogenous variable (0.43), number of predictor (exogenous) variables (5), significance level (0.05), and sample size (370). Based on that, the observed power was calculated to be 1.0, which is acceptable ($> 0.8$) [34–36].

We employed a stratified simple random sampling technique. First, stratification was done based on school ownership into private and governmental schools, which resulted in 5 and 12 schools respectively, and then the sample was allocated proportionally. Then, three governmental (Fasiledess preparatory school, Azezo secondary school, and Hidar 11 secondary school) and two private (Debre Selam St. Mary secondary school and Waliya secondary school) schools were selected on a random basis. Finally, study participants were selected using a simple random sampling technique based on their merged class roaster using Microsoft excel random number generator.

## Study variables

In a multivariate analysis, variables are classified into four categories involving endogenous (independent), exogenous (dependent), latent (unobserved), and observed (manifest) variables. In this regard, the endogenous (dependent) variable of this study was CPB. Whereas, perceived severity, perceived susceptibility, perceived barriers, perceived benefits, self-efficacy, and cues to action were the exogenous (independent) latent variables. A total of 27 manifests (observed) variables were used to measure the latent variables included in the final model. Moreover, sociodemographic attributes of students (age, sex, religion, parental education, parental occupation, family income, living arrangement) were also measured.

## Data collection and measurement

**Data collection.** Data were collected using a structured self-disinterred questionnaire. The self-administered technique was selected over the interviewer-administered method for the following reasons: a) all of the study participants were literate, b) it can reduce social desirability bias, and c) this approach is resource efficient [37, 38]. Four BSc nurses and two public health professionals were participated in the data collection process as a data collector and supervisor respectively after they received a one-day training on the purpose of the study, the

data collection process, the ethical considerations, and the precautions that should be taken during the data collection process. The data were collected at school from Monday to Friday.

The data collectors used hand sanitizers, gloves, and facemasks during the data collection. At the same time, students were also obliged to wear facemasks and clean their hands with hand sanitizer/alcohol. Moreover, each participant was asked for the symptoms of the coronavirus infection by the data collectors before starting any data collection [39].

**Measurements.** The questionnaire used for this study was adapted from different literature by the research team [20, 25, 28, 40, 41]. The instrument was initially prepared in English and then translated into the local language (Amharic). Back translation to English was done to check its consistency. Content validity test and pre-test of the instrument were done based on 10 experts and 21 secondary school students respectively. The content validity was assessed based on six health behavior, two infectious diseases, and two COVID-19 pandemic response team experts' judgment. The final questionnaire used for this study was composed of 51 items with two sections measuring sociodemographic, and HBM variables (perceived severity, perceived susceptibility, perceived barriers, perceived benefits, self-efficacy, cues to action, and CPB).

*Perceived susceptibility*. It was defined as a student's perception of the risk of contracting COVID-19 infection and it was measured by six items having a five-point Likert scale. Its score ranged from 6–30. The higher score indicated higher perceived susceptibility towards COVID-19 [25].

*Perceived severity*. It was defined as a student's perception of the seriousness of having COVID-19 infection and it was measured by 5 items having a five-point Likert scale. Its score ranged from 5–25. The higher score indicated higher perceived severity towards COVID-19 [41, 42].

*Perceived benefit*. It was defined as a student's perception of the benefits of wearing a facemask, keeping physical distance, and washing hands frequently for the prevention of COVID-19 and it was measured by five items having a five-point Likert scale. Its score ranged from 5–25. The higher score indicated higher perceived benefits of performing recommended preventive COVID-19 behaviors [25, 41, 42].

*Perceived barriers*. It was defined as a student's perception of the factors that restrict an individual to do COVID-19 preventive measures and it was measured by four items having a five-point Likert scale. Its score ranged from 4–20. The higher score indicated higher perceived barriers to avoid behavioral risk behaviors of COVID-19 [25].

*Self-efficacy*. It was defined as a student's confidence to execute recommended preventive measures of COVID-19 and it was measured by four items having a five-point Likert scale. Its score ranged from 4 to 20. A higher score indicated the student's higher self-efficacy/confidence to execute the recommended measures [25, 41, 42].

*Cues to action*. It refers to the impact of triggering media, bodily testimonials on student's compliance behavior to the preventive measures of COVID-19. It was measured by three items having a five-point Likert scale. Its score ranged from 3 to 15. The higher score indicates the higher impact of cues to execute preventive behaviors [25, 41, 42].

*Preventive health behaviors*. Refers to the student's practice concerning handwashing, physical distancing, and facemask wearing to prevent COVID-19 infection. It was measured by seven items having a five-point response rate ranging from 1 (Never) to 5 (always). The composite score of the preventive behaviors ranged from 8 to 40. The higher score indicates student's better engagement COVID-19 preventive behaviors [25, 28].

The psychometric properties of each construct are depicted in detail in the result section of this manuscript [Tables 5 and 6].

## Data processing and analysis

The data were entered into EpiData version 4.6 and transferred into STATA version 14 and SMART-PLS version 3.2 statistical software for further data cleaning, coding and analysis. Descriptive statistics such as medians, interquartile ranges, frequencies, and proportions were computed. A structural equation modeling analysis was employed to assess relationships among the latent variables (HBM constructs) and the convergent and discriminate validity of the instrument. Structural equation modeling is a multivariate analytical approach used to simultaneously test and estimate complex causal relationships among variables. It can also be used to assess whether a hypothesized model is consistent with the data collected to reflect a theory [43]. There are two major approaches to structural equation modeling–covariance-based SEM (CB-SEM) and variance-based/partial least squares SEM (PLS-SEM). Even though both approaches are used for the same purpose, to assess cause-effect relations between latent constructs, they differ in their basic assumptions and estimation procedures [44]. PLS-SEM uses a regression-based ordinary least squares (OLS) estimation method intending to explain the latent constructs' variance by minimizing the error terms and maximizing the $R^2$ values of the target endogenous constructs [45]. On the other hand, the CB-SEM estimation procedure aims at reproducing the covariance matrix i.e., minimizing the difference between the observed and estimated covariance matrix, without focusing on explained variance [43]. CB-SEM requires normally distributed data and larger sample size, particularly when the data didn't meet multivariate normality assumption, to produce precise estimates than PLS-SEM which can produce precise estimate with smaller sample size regardless of multivariate normality of the [33, 45, 46]. In the present study, the multivariate normality assumption was assessed and it was markedly departed from the multivariate normality assumption with a Mardia coefficient of 14.8 [47]. Therefore, we preferred to use PLS-SEM since it is an appropriate approach with a smaller sample size regardless of the multivariate assumptions.

The PLS-SEM analysis was done in two steps. In the first step reliability, convergent, and discriminant validity of the instrument were judged based on the assessment of the outer model (a model which shows the relationship between the latent variable and its indicators) by constructing a seven-factor model (initial model) based on the health belief model. The reliability of items within a construct was assessed using Cronbach's coefficient (α) composite reliability of > 0.7 [48].

The discriminant validity was assessed using Hetro-Trait Mono-Trait (HTMT) criterion and all the HBM constructs achieved the discriminant validity, HTMT<0.85 [49]. Whereas, convergent validity was assessed using average variance extracted (AVE), where all HBM constructs except perceived severity achieved (AVE> 0.5) [43]. Moreover, perceived benefit, perceived barrier and CPB achieved convergent validity, AVE > 0.5 following the removal of 1, 3, and 1 poorly loaded items respectively. However, perceived susceptibility failed to achieve construct validity at all because of poor factor loading values of its indicators [Table 5]. Furthermore, the presence of multicollinearity among constructs was assessed using variance inflation factor (VIF). The VIF value of each construct was ranged from 1.43 to 2.95 which laid within the acceptable range, less than five [50].

At the second stage, we constructed a six-factor model (final model) based on the HBM model by excluding perceived susceptibility (because it didn't achieve convergent validity). The bootstrapping procedure was employed to evaluate the structural model empirically and to calculate significant values for all paths [51]. We calculated the amount of variance ($R^2$) in CPB explained by the model and the path coefficients, including the T-value and *P*-value. The $R^2$ criterion value was evaluated based on the previous recommendations: 0.02 as small, 0.13 as a medium, and 0.26 as large. To evaluate our hypotheses, we considered path coefficients with

a T-value >1.96 and a *P*-value <0.05 as significant. Moreover, coefficient of determination (R-Square) and predictive relevance (Q2) were computed through the Blindfolding procedure to assess the final model predictive utility. Accordingly, the final model demonstrated acceptable predictive utility. All PLS-SEM analyses were performed using SmartPLS 3 software.

## Ethical consideration

For this study, ethical clearance was obtained from the Institute Review Board (IRB) of the University of Gondar with an approval number of V/PRCS/05/548/2020. Written consent was obtained from participants aged 18 and above. For participants with the age of less than 18, parental/guardian consent and assent from themselves were obtained. Moreover, permission letters and oral permissions were obtained from the city education office and selected school principals respectively. Each of the participants was included voluntarily and the data were analyzed anonymously. Indeed, the study was conducted following the Declaration of Helsinki [52].

## Result

### Sociodemographic characteristics

A total of 370 students participated in this study with a response rate of 92%. The median age of the participants was 18 with an interquartile age range of 2 years. More than half (51.9%) of the participants were females. The majority of the participants (76.2%) were from private schools (Table 1).

### COVID-19 preventive behaviors

About 80 (21.6%), 62 (16.76%), 81 (21.89) of students reported that they never keep their physical distance, never wash their hands frequently for at least 20 seconds, and never wear facemask respectively. On the other hand, only 97 (26.2%), 121 (32.7%), and 108 (29.2%) of the students reported that they consistently keep their physical distance, wash their hands frequently for at least 20 minutes, and wear facemask respectively [Table 2].

### Health Belief Model variables

The composite score for each construct of HBM was computed by adding indicators value of the same construct. Then, the composite score was divided by the number of indicators for each construct to produce a standardized score to make a comparison across constructs. Only two construct's median score were higher than the neutral score of the Likert scale. The lowest median score was observed in students' perceived severity of the pandemic, whereas the highest score was observed in their perceived benefit of taking preventive measures and cues to action to take preventive measures. The median score of perceived severity was significantly higher among females at a p-value < 0.05. However, there was no significant difference in the median score of all other constructs across gender [Table 3].

   **Correlation among Health Belief Model variable.**   Since the score of most of constructs didn't meet the normality assumption, we employed a Spearman's correlation analysis to assess the relationship between HBM variables [53]. The result revealed that perceived severity (r = 0.2, p < 0.05), perceived benefit (r = 0.27, p < 0,05), self-efficacy (r = 0.52, p < 0,05) and cues to action (r = 0.40, p < 0,05) were positively and significantly correlated with CPB. On the other hand, perceived barrier (r = -0.22, p < 0,05) was negatively correlated with CPB whereas perceived susceptibility didn't show significant correlation with COVID-19 preventive behavior [Table 4].

**Table 1. Socio-demographic characteristics of Gondar city secondary school students, North West Ethiopia, 2020 (n = 370).**

| Variable | Response category | Frequency | Percent |
|---|---|---|---|
| Age | < 18 | 149 | 40.3 |
| | ≥ 18 | 221 | 59.7 |
| Sex | Male | 178 | 48.1 |
| | Female | 192 | 51.9 |
| Marital Status | Single | 328 | 88.7 |
| | Married | 33 | 8.9 |
| | Engaged | 9 | 2.4 |
| Educational status of the participants | Grade 9 | 11 | 3.0 |
| | Grade 10 | 146 | 39.5 |
| | Grade 11 | 93 | 25.1 |
| | Grade 12 | 120 | 32.4 |
| Religion | Orthodox | 331 | 89.5 |
| | Muslim | 39 | 10.5 |
| Mother's occupation | Housewife | 262 | 70.8 |
| | Government employee | 47 | 12.7 |
| | Merchant | 33 | 8.9 |
| | NGO employee | 13 | 3.5 |
| | Other | 15 | 4.1 |
| Father's occupation | Government employee | 92 | 24.9 |
| | NGO employee | 42 | 11.3 |
| | Merchant | 93 | 25.1 |
| | Farmer | 121 | 32.7 |
| | Other | 22 | 6.0 |
| Mother's educational status | Unable to read and write | 131 | 35.4 |
| | Able to read and write | 89 | 24.0 |
| | Primary | 59 | 16.0 |
| | Secondary | 59 | 16.0 |
| | Tertiary | 32 | 8.6 |
| Father's educational status | Unable to read and write | 64 | 17.3 |
| | Able to read and write | 116 | 31.4 |
| | Primary | 59 | 16.0 |
| | Secondary | 70 | 18.9 |
| | Tertiary | 61 | 16.4 |
| To whom do you live? | With my parents | 237 | 64.0 |
| | With my siblings | 49 | 13.2 |
| | With my relatives | 31 | 8.4 |
| | Alone | 43 | 11.6 |
| | Other | 10 | 2.7 |
| School type | Government School | 272 | 76.2 |
| | Private School | 88 | 23.8 |

## Structural equation modeling analysis

Structural equation modeling analysis involves two important model assessments, each has its objectives. These models are measurement model (outer model) assessment and structural model (inner model) assessment. The first one was done to evaluate the psychometric properties of the instrument whereas the second was employed to test the hypothesis that was proposed by the HBM in predicting a CPB.

**Table 2. COVID-19 preventive behaviors among secondary school students in Gondar city, Northwest Ethiopia 2020 (n = 370).**

| Items | Response categories | | | | |
|---|---|---|---|---|---|
| | Poor practice | | | Good practice | |
| | Never | Rarely | Some times | Many times | Always |
| I keep my physical distance | 80 (21.62) | 103 (27.84) | 90 (24.32) | 55 (14.86) | 42 (11.35) |
| I bend my elbow in front of my mouth and nose when I cough or sneeze | 46 (12.43) | 76 (20.54) | 82 (22.16) | 74 (20) | 92 (24.86) |
| I don't shake hands/ kiss others | 64 (17.3) | 120 (32.4) | 92 (24.86) | 41 (11.08) | 53 (14.32) |
| I don't leave the house unless necessary | 92 (24.86) | 87 (23.51) | 99 (26.76) | 48 (12.97) | 44 (11.89) |
| I wash my hands frequently for at least 20 minutes | 62 (16.76) | 83 (22.43) | 104 (28.11) | 68 (18.38) | 53 (14.32) |
| I don't touch my nose, face, and mouth without washing my hands | 76 (20.54) | 66 (17.84) | 83 (22.43) | 60 (16.22) | 85 (22.97) |
| I wear facemask | 81 (21.89) | 86 (23.4) | 95 (25.68) | 57 (15.41) | 51 (13.78) |

**Table 3. Descriptive summary results of Health Belief Model variables.**

| Construct domain | Total | | | Male | | Female | |
|---|---|---|---|---|---|---|---|
| | Score range | Median | IQR | Median | IQR | Median | IQR |
| Perceived susceptibility | 1–5 | 2.9 | 1.8 | 3.2 | 1.6 | 2.8 | 1.7 |
| Perceived severity* | 1–5 | 2.75 | 2 | 2.5 | 1.5 | 3 | 1.25 |
| Perceived benefit | 1–5 | 4 | 1.2 | 4 | 1.2 | 4 | 1 |
| Perceived barrier | 1–5 | 2.85 | 1.6 | 3 | 1.6 | 2.85 | 1.6 |
| Cues to action | 1–5 | 4 | 1.3 | 4 | 1 | 4 | 1.3 |
| Self-efficacy [n] | 1–5 | 3 | 1.75 | 3 | 0.93 | 2.99 | 1.03 |
| CPB | 1–5 | 2.88 | 1.4 | 2.75 | 1.37 | 2.94 | 1.4 |

* Shown significant difference across gender at p-value <0.05, CPB = COVID-19 Preventive Behavior

[n] Mean and standard deviation are reported instead of median and interquartile ranges respectively because the construct's score was normally distributed,

IQR = Interquartile range.

**Table 4. Spearman correlation among Health Belief Model variables.**

| Constructs | PSU | PSE | PBE | PBA | CA | SEF | CPB |
|---|---|---|---|---|---|---|---|
| PSU | 1.00 | | | | | | |
| PSE | 0.26 (<0.001) | 1.00 | | | | | |
| PBE | 0.13 (0.02) | 0.34 (<0.001) | 1.00 | | | | |
| PBA | 0.08(0.62) | 0.04 (0.28) | -0.12 (0.06) | 1.00 | | | |
| CA | 0.14(0.01) | 0.23(<0.001) | 0.44 (<0.001) | -0.20(0.01) | 1.00 | | |
| SEF | 0.09(0.07) | 0.15(0.01) | 0.29 (<0.001) | -0.12(0.01) | 0.48(<0.001) | 1.00 | |
| CPB | 0.10 (0.11) | 0.20(<0.001) | 0.27(<0.001) | -0.22(0.002) | 0.40(<0.001) | 0.52(<0.001) | 1.00 |

Note: values in the bracket in each cell represents a p-value and values out of the bracket are correlation coefficients (r), PSU = Perceived susceptibility, PSE = Perceived severity, PBE = Perceived benefit, PBA = Perceived barrier, CA = Cues to action, SEF = Self-efficacy, and CPB = COVID-19 preventive behavior.

Concerning the measurement model assessment, all of the HBM constructs shown adequate reliability and convergent validity except perceived susceptibility. The results indicated that all of the HBM model constructs were measured adequately, where each of the constructs captured 50% of the variance in its indicators [Table 5].

**Discriminant validity.** Discriminant validity ensures that a constructed measure is empirically unique and represents phenomena of interest that other measures in a structural

**Table 5. The reliability and convergent validity tests results of the instrument used to assess students CPB based on the Health Belief Model in Gondar city, Northwest Ethiopia, 2020 (n = 370).**

| Construct domain | Indicator | Initial Model | | | | Final Model | | | |
|---|---|---|---|---|---|---|---|---|---|
| | | loading | Alpha | CR | AVE | Loading | Alpha | CR | AVE |
| Cues to action | CA1 | 0.66 | 0.76 | 0.77 | 0.52 | 0.66 | 0.76 | 0.78 | 0.52 |
| | CA2 | 0.83 | | | | 0.84 | | | |
| | CA3 | 0.66 | | | | 0.66 | | | |
| CPB | CPB1 | 0.79 | 0.89 | 0.89 | 0.51 | 0.78 | 0.88 | 0.88 | 0.52 |
| | CPB2 | 0.67 | | | | 0.65 | | | |
| | CPB3 | 0.72 | | | | 0.70 | | | |
| | CPB4 | 0.71 | | | | 0.69 | | | |
| | CPB5 | 0.74 | | | | 0.73 | | | |
| | CPB6 | 0.55 | | | | Omitted | | | |
| | CPB7 | 0.74 | | | | 0.73 | | | |
| | CPB8 | 0.75 | | | | 0.74 | | | |
| Perceived barrier | PBA1 | 0.82 | 0.86 | 0.86 | 0.62 | 0.74 | 0.84 | 0.85 | 0.56 |
| | PBA2 | 0.76 | | | | 0.69 | | | |
| | PBA3 | 0.93 | | | | 0.89 | | | |
| | PBA4 | 0.51 | | | | Omitted | | | |
| | PBA5 | 0.47 | | | | Omitted | | | |
| | PBA6 | 0.24 | | | | Omitted | | | |
| | PBA7 | 0.71 | | | | 0.65 | | | |
| Perceived benefit | PBE1 | 0.79 | 0.87 | 0.85 | 0.48 | 0.76 | 0.86 | 0.86 | 0.54 |
| | PBE2 | 0.80 | | | | 0.77 | | | |
| | PBE3 | 0.74 | | | | 0.68 | | | |
| | PBE4 | 0.76 | | | | 0.72 | | | |
| | BEN 5 | 0.60 | | | | Omitted | | | |
| | PBE6 | 0.76 | | | | 0.75 | | | |
| Perceived severity | PSE1 | 0.78 | | | | 0.79 | 0.86 | 0.87 | 0.62 |
| | PSE2 | 0.78 | | | | 0.78 | | | |
| | PSE3 | 0.68 | | | | 0.67 | | | |
| | PSE4 | 0.89 | | | | 0.89 | | | |
| Self-efficacy | SEF1 | 0.72 | 0.82 | 0.82 | 0.54 | 0.72 | 0.82 | 0.83 | 0.53 |
| | SEF2 | 0.67 | | | | 0.67 | | | |
| | SEF3 | 0.69 | | | | 0.70 | | | |
| | SEF4 | 0.84 | | | | 0.83 | | | |
| Perceived susceptibility | PSU1 | 0.52 | 0.78 | 0.72 | 0.42 | | | | |
| | PSU2 | 0.82 | | | | | | | |
| | PSU3 | 0.77 | | | | | | | |
| | PSU4 | 0.87 | | | | | | | |
| | PSU5 | -0.30 | | | | | | | |
| | PSU6 | 0.34 | | | | | | | |

Note: AVE = Average Variance Extracted, CR = Composite Reliability, CA = Cues to action, CBM = COVID-19 preventive behaviors, PBA = Perceived barrier, PBE = Perceived benefit, PSE = Perceived severity, SEF = Self efficacy, "Initial model" is a seven-factor with all HBM variables, "Final model" is a six-factor model with all HBM variables except perceived susceptibility.

equation model do not capture [54]. This was assessed by using heterotrait-monotrait ratio of correlations (HTMT); a new method for assessing discriminant validity in partial least squares structural equation modeling [49]. The HTMT value of 0.85 was used to declare convergent

**Table 6. Heterotrait-Monotrait (HTMT) discriminant validity test result of the instrument used to assess students COVID-19 preventive behavior based on the Health Belief Model in Gondar city, Northwest Ethiopia, 2020 (n = 370).**

| Construct Domains | CPB | CA | PBA | PBE | PSE | SEF |
|---|---|---|---|---|---|---|
| COVID-19 preventive behavior (CPB) | | | | | | |
| Cues to action (CA) | 0.49 | | | | | |
| Perceived barrier (PBA) | 0.25 | 0.25 | | | | |
| Perceived benefit (PBE) | 0.31 | 0.55 | 0.15 | | | |
| Perceived susceptibility (PSE) | 0.23 | 0.27 | 0.06 | 0.40 | | |
| Self-Efficacy (SEF) | 0.62 | 0.61 | 0.14 | 0.34 | 0.18 | |

validity. As it is depicted in the table below all of the HBM constructs showed acceptable discriminant validity [Table 6].

**Structural model assessment.** The structural model assessment was done to test the hypothesis proposed by the health belief model. In this regard, a six-factor model was fitted to assess the associations among the HBM variables. In the final model, perceived severity, perceived benefit, perceived barrier, self-efficacy, and cues to action were fitted as exogenous latent variables to predict the endogenous (independent) variable (CPB) as proposed by the HBM. The final model explained 43% of the variance in CPB, indicated that the model showed an adequate predictive utility [Fig 1].

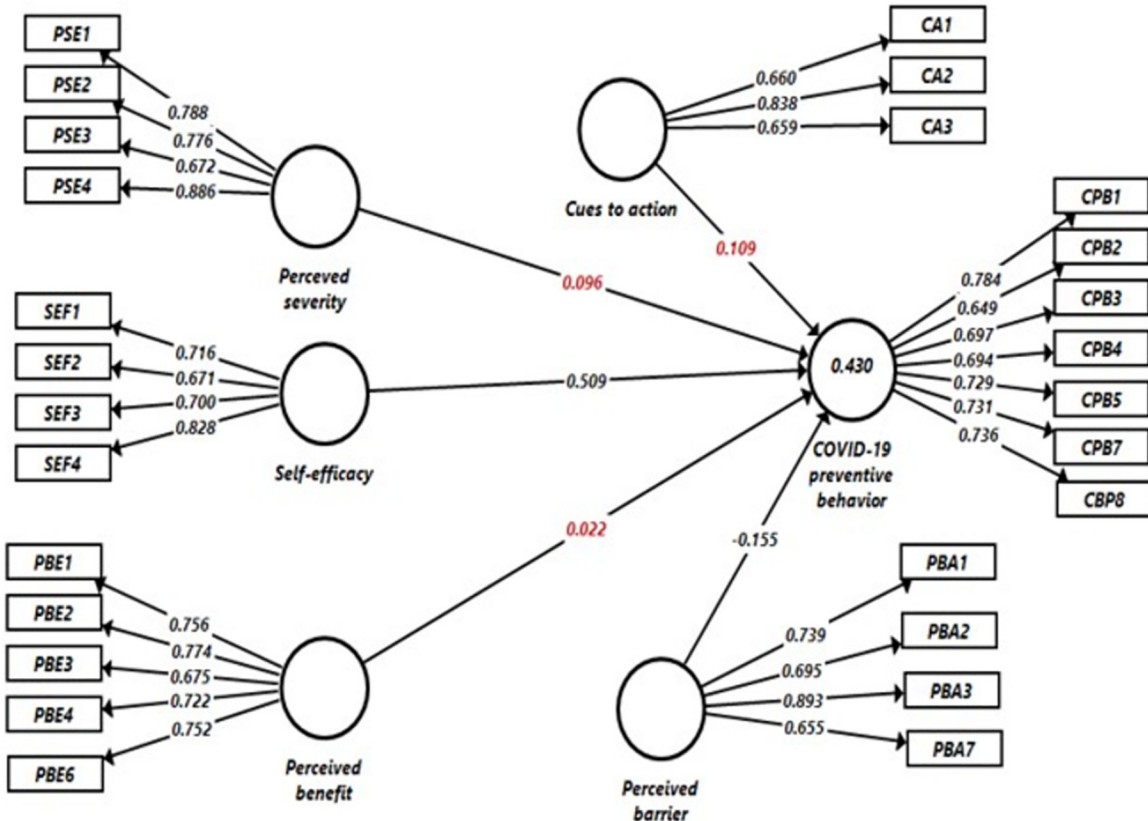

**Fig 1. Structural equation model of determinants of CPB among secondary school students in Gondar city, Northwest Ethiopia, 2020.** CA = Cues to action, CBM = COVID-19 preventive behaviors, PBA = Perceived barrier, PBE = Perceived benefit, PSE = Perceived severity, SEF = Self efficacy, Path coefficients shown in red color were not statistically significant.

**Table 7. Path coefficient of the structural equation modeling analysis of CPB among secondary school students in Gondar city, Northwest Ethiopia, 2020 (n = 370).**

| Hypothesis | Path coefficients | 95% CI | | T-Value | P Values |
|---|---|---|---|---|---|
| | | Lower Bound | Upper bound | | |
| Cues to action → CPB | 0.11 | -0.04 | 0.27 | 1.29 | 0.20 |
| Perceived barrier → CPB | -0.15 | -0.24 | -0.06 | 3.17 | <0.001 |
| Perceived benefit → CPB | 0.02 | -0.13 | 0.15 | 0.30 | 0.77 |
| Perceived severity → CPB | 0.10 | -0.02 | 0.20 | 1.76 | 0.08 |
| Self-efficacy → CPB | 0.51 | 0.36 | 0.64 | 7.19 | <0.001 |

$R^2$ = 0.43, CBM COVID-19 Preventive Behavior.

The structural equation modeling analysis revealed that perceived barriers ($\beta$ = - 0.15, p = <0.001) and self-efficacy to execute the recommended COVID-19 measures ($\beta$ = 0.51, p = <0.001) were significantly linked to CPB positively and negatively respectively. This indicated that students were more likely to engage in COVID-19 preventive behaviors if their perception of barriers to take the recommended measures was lower or their confidence to execute those recommended measures were high. Perceived benefit, perceived severity, and cues to action were linked positively to CPB. However, none of them were statistically significant. Moreover, self-efficacy was a powerful predictor of CPB [Table 7].

## Discussion

This research uses the health belief model to assess predictors of COVID-19 preventive behavior (CPB) of secondary school students in Gondar city. The model explained a 43% variance in the CPB of students. This indicates the model was adequate in predicting the CPB and it may be used to guide behavior change interventions among students in the study area [55]. The result of this study is higher than that of a study conducted in Iran [26] and lower than that of a study conducted in Egypt [42], where the HBM explained 26% and 58% of the variance in CPB respectively. This difference may be due to the different methods of analysis between the current and previous studies. Previous studies have used ordinary regression analysis that may affect the predictive utility of the model because these analysis approaches don't account for measurement errors. In contrast, this study used a multivariate analysis approach that provides a precise estimate against ordinary regression analyses because it takes measurement errors into account.

Only 97 (26.2%), 121 (32.7%), and 108 (29.2%) of students had good practice regarding physical distance, frequent hand washing, and facemask use respectively. These results are lower than the findings reported by different previous studies [15, 16, 26, 29]. The discrepancy may be explained by the fact that previous studies were conducted at the beginning of the introduction of COVID-19 when everybody was scared and had taken preventive measures aggressively.

As time goes by, the risk perception of individuals may be reduced because they have received a great deal of information about the nature of the pandemic. As a result, people's engagement in preventive practices may be reduced. Moreover, the results indicate that most students were not performing the recommended preventive measures for COVID-19. This requires urgent measures, such as school health communications on COVID-19 prevention, to improve students' COVID-19 prevention practice. Besides, students come to school from different villages, which can increase the risk of pandemic transmission in their communities if they become infected. As such, there is a need to encourage students to follow recommended COVID-19 prevention measures for their benefit and the benefit of their community as well.

Among the HBM constructs, perceived barriers and self-efficacy were found to be significant predictors of CPB in the present study. Various HBM -studies conducted based on

the HBM have also identified perceived barriers as a significant determinant of poor adherence to CPB [26, 28, 29, 42]. In our study, perceived barriers had a significant negative association with COVID-19 preventive behavior. This indicated that students were more likely to adopt COVID-19 preventive behaviors if their perceived barriers (lack of soap, lack of sanitizers/alcohol, the impact of COVID-19 preventive measures on daily activities, and poor economic status) were eliminated. In addition to this, our PLS-SEM analysis also revealed that self-efficacy was another significant predictor of COVID-19 preventive behaviors. This result indicated that student engagement with COVID-19 preventive behavior was dependent on their perceived self-efficacy/confidence to take the recommended measures. These results are consistent with various studies conducted previously elsewhere based on the health belief model [26, 28, 29, 40–42]. In the present study, perceived self-efficacy was the most powerful predictor of students' COVID-19 preventive behaviors, identified the need to focus on improving student self-efficacy (confidence) to adopt COVID-19 preventative behavior for students to follow recommended actions. This finding is contradictory with a systematic review of HBM-based studies that claimed that perceived barrier was the strongest predictor of preventive health behaviors [30]. However, it is consistent with some other studies that were done using the health belief model to predict COVID-19 preventive behaviors [28, 40, 41].

In the present study, perceived benefit, perceived severity, and cues to action showed a positive correlation with the COVID-19 preventive behaviors as proposed by the HBM. However, none of them showed any significant association in the structural equation modeling analysis. These findings were contradictory in other studies. On the other hand, the results are consistent with various previous studies based on HBM that indicated that perceived severity was not an important predictor of COVID-19-related preventative behaviors, where perceived benefit [26, 29] and cues to action [28] were significant predictors of COVID-19 preventive behaviors. On the other hand, the results of this study are consistent with various previous HBM based studies which claimed that perceived severity was not a significant predictor of COVID-19 preventive behaviors [26, 28, 42].

The present study has several limitations including: it was solely based on self-reported responses of students that may be liable to social desirability bias, and the study was based on the intrapersonal level model, (HBM) where other environmental and interpersonal factors were not considered. Withstanding the aforementioned limitation, the study applied a structural equation modeling analysis which is supposed to produce a precise estimate in analyses involving latent variables, because this analysis technique takes measurement errors into account unlike the ordinary regression analysis [56].

## Conclusion and recommendations

COVID-19 prevention practice is quite low among students. The Health Belief Model demonstrated adequate predictive utility in predicting preventive behaviors related to COVID-19. Perceived barrier and self-efficacy were found to be the significant predictors of COVID-19 preventive behavior. Thus, to contain the spread of the COVID-19 virus, schools should design and implement behavioral change programs to enhance protective behavior amongst students, by issuing warnings and recommendations about the pandemic, or by imposing legal restrictions accordingly. Moreover, such interventions should focus on the reduction of barriers (providing facemasks, hand sanitizers, and soaps for those who are in financial hardship) and enhancing students' self-efficacy in adopting COVID-19 preventative behavior. Furthermore, the health belief model can also be used to guide behavior change interventions and other researches on the area of interest in the study area.

The authors believed that the present study generated a shred of preliminary evidence for COVID-19 prevention programs in schools in the study area. As we observed, most students have not implemented the recommended COVID-19 prevention measures. This could also lead to an increased risk of COVID-19 transmission in the community if they get infected, as long as students are expected to return to their homes. From this perspective, designing and implementing school-based behavioral change programs to enable students to follow recommended preventive measures will be of direct importance to the entire community.

## Supporting information

**S1 Fig. Conceptual framework adapted based on the Health Belief Model and review of different literatures [20, 26, 28, 40, 41].**
(TIF)

## Acknowledgments

We would like to forward our heartfelt gratitude to the University of Gondar for providing us ethical clearance to do this practical research. We would like to acknowledge study participants, data collectors, and the Gondar city education office for their time and contribution to this work.

## Author Contributions

**Conceptualization:** Kegnie Shitu, Asmamaw Adugna, Ayenew Kassie, Simegnew Handebo.

**Data curation:** Kegnie Shitu.

**Formal analysis:** Kegnie Shitu.

**Investigation:** Kegnie Shitu, Asmamaw Adugna, Ayenew Kassie, Simegnew Handebo.

**Methodology:** Kegnie Shitu, Asmamaw Adugna, Ayenew Kassie, Simegnew Handebo.

**Project administration:** Kegnie Shitu, Asmamaw Adugna, Ayenew Kassie, Simegnew Handebo.

**Resources:** Kegnie Shitu, Asmamaw Adugna, Ayenew Kassie, Simegnew Handebo.

**Software:** Kegnie Shitu.

**Supervision:** Kegnie Shitu, Asmamaw Adugna, Ayenew Kassie, Simegnew Handebo.

**Validation:** Kegnie Shitu, Asmamaw Adugna, Ayenew Kassie, Simegnew Handebo.

**Visualization:** Kegnie Shitu.

**Writing – original draft:** Kegnie Shitu.

**Writing – review & editing:** Kegnie Shitu, Asmamaw Adugna, Ayenew Kassie, Simegnew Handebo.

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
