## [Decision Letter · Decision Letter 0]

17 Mar 2021

PONE-D-21-05732

Application of Health Belief Model for the Assessment of COVID-19 Preventive Behavior and its Determinants among Students: A Structural Equation Modeling Analysis

PLOS ONE

Dear Dr. Shitu,

Thank you for submitting your manuscript to PLOS ONE. After careful consideration, we feel that it has merit but does not fully meet PLOS ONE’s publication criteria as it currently stands. Therefore, we invite you to submit a revised version of the manuscript that addresses the points raised during the review process.

On receipt and processing, your manuscript was sent to two reviewers for assessment. I am now in receipt of the recommendation from the two reviewers, which you will find at the bottom of this email. I have also now had the opportunity to re-read the manuscript myself, and I have two additional observations which I wish to offer in conjunction with the reviewers’ reports.

Your work ignores a substantial research body where COVID-19/Pandemic/wartime perceptions were conceptualised, and this work needs to be consulted with. See Mahmoud et al. [1] and Mahmoud and Reisel [2]Please, whilst revising, offer your manuscript proper proofreading.

References

1.            Mahmoud AB, Grigoriou N, Fuxman L, Reisel WD, Hack-Polay D, Mohr I: **A generational study of employees’ customer orientation: a motivational viewpoint in pandemic time**. *Journal of Strategic Marketing *2020:1-18.

2.            Mahmoud AB, Reisel WD: **Exploring Personal Experience of Wartime Crisis Effects on Job Insecurity in Syria**. *Psihologia Resurselor Umane *2015, **13**(2):245–256.

We look forward to receiving your revised manuscript.

Kind regards,

Ali B. Mahmoud, Ph.D.

Academic Editor

PLOS ONE

Please provide additional details regarding participant consent. In the ethics statement in the Methods and online submission information, please ensure that you have specified whether consent was informed.

Please include your actual numerical p-values in Table 4.

Please revise your tables to replace p-values of "0.000" to "<0.001".

Please provide the names of the five schools were participants were recruited from.

Please include additional information regarding the survey or questionnaire used in the study and ensure that you have provided sufficient details that others could replicate the analyses. For instance, if you developed the survey or questionnaire as part of this study and it is not under a copyright more restrictive than CC-BY, please include a copy, in both the original language and English, as Supporting Information. If the questionnaire is published, please provide a citation to the (1) questionnaire and/or (2) original publication associated with the questionnaire.

7. We note that you have indicated that data from this study are available upon request. PLOS only allows data to be available upon request if there are legal or ethical restrictions on sharing data publicly. For information on unacceptable data access restrictions, please see http://journals.plos.org/plosone/s/data-availability#loc-unacceptable-data-access-restrictions.

8. We note you have included a table to which you do not refer in the text of your manuscript. Please ensure that you refer to Table 1 in your text; if accepted, production will need this reference to link the reader to the Table.

Reviewers' comments:

Reviewer's Responses to Questions

**Comments to the Author**

1. Is the manuscript technically sound, and do the data support the conclusions?

Reviewer #1: Yes

Reviewer #2: Yes

2. Has the statistical analysis been performed appropriately and rigorously? 

Reviewer #1: Yes

Reviewer #2: Yes

3. Have the authors made all data underlying the findings in their manuscript fully available?

Reviewer #1: No

Reviewer #2: Yes

4. Is the manuscript presented in an intelligible fashion and written in standard English?

Reviewer #1: No

Reviewer #2: Yes

5. Review Comments to the Author

Reviewer #1: The strongest contribution of this study is the findings. This is because the results show that students in the Ethiopian city of Gondar do not have good practice of Covid-19 preventive measures. Methodology is sound and well executed. However, post-hoc power analysis should be conducted to ascertain statistical power for sample size adequacy (Refer to lines 123 – 130). The conclusion is logically derived from the analysis of data. However, this reviewer recommends the author/s to improve on the practical recommendations based on the findings of the study. Some thought should be given to what measures should be in place to improve the adoption of good practices of Covid-19 preventive measures among students. This is the weakest part of the article. Some typos (in the Table as well) exist in the manuscript. For example, expression like “pandemic that eats many lives" (line 66). This article requires editorial work to improve the written quality. Furthermore, the referencing style needs to conform to the standard expected by the Journal.

Reviewer #2: The effort invested in the crafting of this paper is apparent. However, the paper in its current form needs the suggested improvements in order to overcome the listed limitations and produce a stronger contribution to the state of existing knowledge within a reasonable timeframe. As a result, I am recommending a resubmit addressing the recommended changes for this paper prior to its acceptance in journal. I hope that the review comments will help further improve the work, or reframe it in such a way that a revised submission would be strengthened.

(1) In the abstract and introduction section, the rationale for and the aim/motivation of the study has not been stated, the rationale (academic underpinning) for conducting this study is not strong and is currently weak. E.g. What is the originality of this study?

(2) The paper provides a satisfactory literature review demonstrating an understanding of the relevant literature in the field with the appropriate up-to-date literature sources.

(3) Although the research methodology applied for the study is appropriate, this section lacks academic justifications to support your methodology choices – some of the statements/claims made sounds more arbitrary. There is a need to discuss the alternatives and justification of methods selected for data collection and data analysis, supported by appropriate references needs to be provided.

(4) An interesting area of research and the results provide significant theoretical and practical contributions but the discussions section could better relate the findings to entries information management applications.

(5) The current conclusion section needs to answer the following questions:

• How does the paper bridge the gap between theory and practice?

• What is the impact upon society?

(6) Overall, paper is well-written and structured, which does much for its readability and comprehensibility.

Best Wishes for revision

6. PLOS authors have the option to publish the peer review history of their article (what does this mean?). If published, this will include your full peer review and any attached files.

Reviewer #1: No

Reviewer #2: No

---

## [Author Response · Author response to Decision Letter 0]

10 Apr 2021

Dear Ali B. Mahmoud, Ph.D., editor and reviewers:

Hereby, we resubmit the enclosed –revised- manuscript ID PONE-D-21-05732-[ EMID:15744fc9849b3e3d], which is entitled “Application of Health Belief Model for the Assessment of COVID-19 Preventive Behavior and its Determinants among Students: A Structural Equation Modeling Analysis” to your journal.

We have read the comments carefully and we were able to implement all of them (see the manuscript with track changes). While reading the manuscript critically, we spotted English grammar errors, which we and a language expert have corrected too. We hope that our revision will be felt like an improvement. We certainly feel this manuscript has improved thanks to the suggestions of the reviewers. The datasets generated and/or analyzed in the current study are available publicly with no restriction at https://www.kaggle.com/kegnieshitu/determinants-of-covid19-preventive-behaviors

Response to the Editor’s Comment and suggestions

On receipt and processing, your manuscript was sent to two reviewers for assessment. I am now in receipt of the recommendation from the two reviewers, which you will find at the bottom of this email. I have also now had the opportunity to re-read the manuscript myself, and I have two additional observations which I wish to offer in conjunction with the reviewers’ reports.

Your work ignores a substantial research body where COVID-19/Pandemic/wartime perceptions were conceptualised, and this work needs to be consulted with. See Mahmoud et al. [1] and Mahmoud and Reisel [2]

Please, whilst revising, offer your manuscript proper proofreading.

References

1. Mahmoud AB, Grigoriou N, Fuxman L, Reisel WD, Hack-Polay D, Mohr I: A generational study of employees’ customer orientation: a motivational viewpoint in pandemic time. Journal of Strategic Marketing 2020:1-18.

2. Mahmoud AB, Reisel WD: Exploring Personal Experience of Wartime Crisis Effects on Job Insecurity in Syria. Psihologia Resurselor Umane 2015, 13(2):245–256.

Response: Thank you very much, Dr. Ali B. Mahmoud, for your input and comments. I looked at two of the documents and found them both interesting. I tried to incorporate the findings of the first article in the background section of our manuscript (See page 5, line 93-97 in the manuscript). However, the second article is not related to the present study. While its primary focus is not related to COVID-19, the paper attempted to provide an overview of how job insecurity was in the Syrian war crisis. On the other hand, the present study attempted to provide an overview of how students are implementing the recommended preventive actions for COVID-19. Because of this, we could not consult the second article.

Response to Journal requirements:

Please ensure that your manuscript meets PLOS ONE's style requirements, including those for file naming. The PLOS ONE style templates can be found athttps://journals.plos.org/plosone/s/file?id=wjVg/PLOSOne_formatting_sample_main_body.pdf andhttps://journals.plos.org/plosone/s/file?id=ba62/PLOSOne_formatting_sample_title_authors_affiliations.pdf

Response: We have prepared the manuscript in accordance with the guidelines presented at the aforementioned web addresses..

Please provide additional details regarding participant consent. In the ethics statement in the Methods and online submission information, please ensure that you have specified whether consent was informed.

Response: We referred to the ethics statement and stated that we have obtained ethical approval from the University of Gondar. As well, we obtained consent from each participant >=18 years of age and participants <18 years of age from their parents/guardians and their assent (See page 14, line 292-300 in the manuscript).

Please include your actual numerical p-values in Table 4.

Response: P-values are included for each correlation coefficient (See table 4 on page 19 in manuscript) 

Please revise your tables to replace p-values of "0.000" to "<0.001".

Response: Correction was made accordingly (see Table 7 on page 23 in the manuscript)

Please provide the names of the five schools were participants were recruited from.

Response: We described the list of selected schools (See line 169-171 on page 8 in the manuscript)

Please include additional information regarding the survey or questionnaire used in the study and ensure that you have provided sufficient details that others could replicate the analyses. For instance, if you developed the survey or questionnaire as part of this study and it is not under a copyright more restrictive than CC-BY, please include a copy, in both the original language and English, as Supporting Information. If the questionnaire is published, please provide a citation to the (1) questionnaire and/or (2) original publication associated with the questionnaire.

Response: The tool was prepared by adapting instruments from various studies and that we provided an appropriate citation to the studies associated with the questionnaire. (See line 197-198 on page 10 in the manuscript)

7. We note that you have indicated that data from this study are available upon request. PLOS only allows data to be available upon request if there are legal or ethical restrictions on sharing data publicly. For information on unacceptable data access restrictions, please see http://journals.plos.org/plosone/s/data-availability#loc-unacceptable-data-access-restrictions.

Response: We made our dataset publicly available at:

https://www.kaggle.com/kegnieshitu/determinants-of-covid19-preventive-behaviors

8. We note you have included a table to which you do not refer in the text of your manuscript. Please ensure that you refer to Table 1 in your text; if accepted, production will need this reference to link the reader to the Table.

Response: Sorry for the mistake. We cited the table in its appropriate place (See line 304 on page 14 in the manuscript)

Response to reviewer 1 comments and suggestions

The strongest contribution of this study is the findings. This is because the results show that students in the Ethiopian city of Gondar do not have good practice of Covid-19 preventive measures. Methodology is sound and well executed. However, post-hoc power analysis should be conducted to ascertain statistical power for sample size adequacy (Refer to lines 123 – 130). The conclusion is logically derived from the analysis of data. However, this reviewer recommends the author/s to improve on the practical recommendations based on the findings of the study. Some thought should be given to what measures should be in place to improve the adoption of good practices of Covid-19 preventive measures among students. This is the weakest part of the article. Some typos (in the Table as well) exist in the manuscript. For example, expression like “pandemic that eats many lives" (line 66). This article requires editorial work to improve the written quality. Furthermore, the referencing style needs to conform to the standard expected by the Journal.

Response: Thank you for your comment. We did a post hoc power analysis based on assumptions including, the amount of examining variable in the endogenous variable (COVID-19 preventive behavior=0.43), the number of predictors (exogenous variables=5), significant level=0.05, and sample size=370. The computation was done using an online statistical calculator which was developed by an American professor, Daniel Soper (1) (See line 160-164 on page 8 in the manuscript).

1. Soper, D.S. (2021). Post-hoc Statistical Power Calculator for Multiple Regression [Software]. Available from https://www.danielsoper.com/statcalc

Moreover, we gave this manuscript to an English language expert, Mr. Fikadie, for an English language edition. Then the entire manuscript was edited for grammatical and typographic errors based on his suggestion and comments. We checked our referencing style whether it is per the journal requirement and made corrections. In addition to this, we tried to correct contextual errors we had noted in our proofreading and emotional expressions like “pandemic that eats many lives” were revised (see line 75-76 on page 4 in the manuscript). 

Response to reviewer 2 comments and suggestions

Reviewer #2: The effort invested in the crafting of this paper is apparent. However, the paper in its current form needs the suggested improvements to overcome the listed limitations and produce a stronger contribution to the state of existing knowledge within a reasonable timeframe. As a result, I am recommending a resubmit addressing the recommended changes for this paper prior to its acceptance in journal. I hope that the review comments will help further improve the work, or reframe it in such a way that a revised submission would be strengthened.

(1) In the abstract and introduction section, the rationale for and the aim/motivation of the study has not been stated, the rationale (academic underpinning) for conducting this study is not strong and is currently weak. E.g. What is the originality of this study?

Response: Thank you for your valuable suggestions and comments. We tried to further describe the rationale and its originality in the introduction sections, (see line 113-132 on page 6 in the manuscript)

(2) The paper provides a satisfactory literature review demonstrating an understanding of the relevant literature in the field with the appropriate up-to-date literature sources.

Response: Thank you for your feedback. 

 (3) Although the research methodology applied for the study is appropriate, this section lacks academic justifications to support your methodology choices – some of the statements/claims made sounds more arbitrary. There is a need to discuss the alternatives and justification of methods selected for data collection and data analysis, supported by appropriate references need to be provided.

Response: We discussed the justifications for the data collection and analysis methodology we had used in this study. For instance, we followed a self-administered approach to collect our data for the following reasons; a) all of our participants were literates, b) the self-administered technique is better in reducing social desirability bias over the interviewer-administered approach, and c) the self-administered technique saves time and resource (see in line 185-188 at page 9 in the manuscript)

Regarding the analysis technique, we employed a structural equation modeling analysis which has been widely used by health behavior researchers investigating complex relationships between latent constructs such as perceived susceptibility, COVID-19 preventive behaviors, and so on. There are two approaches of structural equation modeling approaches; covariance-based SEM (CB-SEM) and variance-based same the so-called partial least squares SEM(PLS-SEM). Even though these approaches have the same purpose, they differ in their basic assumptions. The former is based on the maximum likelihood estimation technique that requires data that meets the multivariate normality assumption. Whereas the latter computes estimates using ordinary least squares estimation techniques regardless normality of the data. Moreover, the PLS-SEM is effective over CB-SEM with a small sample size and/or data that violate multivariate assumptions. In our case, we preferred to use PLS-SEM because our data didn’t meet the multivariate normality assumption (the details of comparison of these methods and supporting articles are presented in the manuscript, see in line 245-363 on page 12).

(4) An interesting area of research and the results provide significant theoretical and practical contributions but the discussions section could better relate the findings to entries information management applications.

Response: We made revisions to the discussion part by further discussing the impact of the findings on actual practice. (see the discussion part in the revised manuscript)

(5) The current conclusion section needs to answer the following questions:

• How does the paper bridge the gap between theory and practice?

• What is the impact upon society?

Response: We tried to mention how the current study findings could brig the gap b/n the theory and practice and its implication of on the society (See the conclusion part in the revised manuscript and see line 455-471 from page 26-27 in the manuscript)

(6) Overall, the paper is well-written and structured, which does much for its readability and comprehensibility.

Response: Thank you so much!

---

## [Decision Letter · Decision Letter 1]

15 Apr 2021

PONE-D-21-05732R1

Application of Health Belief Model for the Assessment of COVID-19 Preventive Behavior and its Determinants among Students: A Structural Equation Modeling Analysis

PLOS ONE

Dear Dr. Shitu,

Thank you for submitting your manuscript to PLOS ONE. After careful consideration, we feel that it has merit but does not fully meet PLOS ONE’s publication criteria as it currently stands. Therefore, we invite you to submit a revised version of the manuscript that addresses the points raised during the review process.

Outstanding from the previous review: AB Mahmoud and WD Reisel [1] conceptualise wartime perceptions which can form the theoretical premise for COVID-19 perceptions since the latter has triggered a war-like context— therefore, it has to be cited. While revising and addressing Reviewer 1 comments, make sure an English native speaker proofreads the whole text.

References

1.            Mahmoud AB, Reisel WD: **Exploring Personal Experience of Wartime Crisis Effects on Job Insecurity in Syria**. *Psihologia Resurselor Umane *2015, **13**(2):245–256.

We look forward to receiving your revised manuscript.

Kind regards,

Ali B. Mahmoud, Ph.D.

Academic Editor

PLOS ONE

Reviewers' comments:

Reviewer's Responses to Questions

**Comments to the Author**

1. If the authors have adequately addressed your comments raised in a previous round of review and you feel that this manuscript is now acceptable for publication, you may indicate that here to bypass the “Comments to the Author” section, enter your conflict of interest statement in the “Confidential to Editor” section, and submit your "Accept" recommendation.

Reviewer #1: All comments have been addressed

Reviewer #2: All comments have been addressed

2. Is the manuscript technically sound, and do the data support the conclusions?

Reviewer #1: Yes

Reviewer #2: Yes

3. Has the statistical analysis been performed appropriately and rigorously? 

Reviewer #1: Yes

Reviewer #2: Yes

4. Have the authors made all data underlying the findings in their manuscript fully available?

Reviewer #1: Yes

Reviewer #2: Yes

5. Is the manuscript presented in an intelligible fashion and written in standard English?

Reviewer #1: No

Reviewer #2: Yes

6. Review Comments to the Author

Reviewer #1: In my first review of this article, I made comments on the need for post-hoc power analysis. This was included. Furthermore, I commended the authors of the important contribution of the article related to children and Covid-19 prevention behaviors and recommended authors to add relevant practical recommendations. What is unclear is the phrase "behavior change barograms" (line 488). Lastly, this revised submission still requires professional proofreading and editing to improve English expressions and grammar.

Reviewer #2: I am pleased to see that that the authors have taken the previous review comments on board and have set out to deal with every one of them. There has clearly been a great deal of additional work undertaken to address each of the reviewers' concerns, which is duly noted and appreciated. This has made a significant impact on the quality of the paper. Thank you.

7. PLOS authors have the option to publish the peer review history of their article (what does this mean?). If published, this will include your full peer review and any attached files.

Reviewer #1: **Yes: **Dr Seong-Yuen Toh

Reviewer #2: No

---

## [Author Response · Author response to Decision Letter 1]

26 Apr 2021

Dear Ali B. Mahmoud, Ph.D., editor and reviewers:

Hereby, we resubmit the enclosed –revised- manuscript ID PONE-D-21-05732-[ EMID:15744fc9849b3e3d], which is entitled “Application of Health Belief Model for the Assessment of COVID-19 Preventive Behavior and its Determinants among Students: A Structural Equation Modeling Analysis” to your journal.

We have read the comments carefully and we were able to implement all of them (see the manuscript with track changes). While reading the manuscript critically, we spotted English grammar errors, which we and a language expert have corrected too. We hope that our revision will be felt like an improvement. We certainly feel this manuscript has improved thanks to the suggestions of the reviewers. 

Response to the Editor’s Comment and suggestions

Outstanding from the previous review: AB Mahmoud and WD Reisel [1] conceptualise wartime perceptions which can form the theoretical premise for COVID-19 perceptions since the latter has triggered a war-like context— therefore, it has to be cited. While revising and addressing Reviewer 1 comments, make sure an English native speaker proofreads the whole text.

References

1. Mahmoud AB, Reisel WD: Exploring Personal Experience of Wartime Crisis Effects on Job Insecurity in Syria. Psihologia Resurselor Umane 2015, 13(2):245–256.

Response: Thank you very much, Dr. Ali B. Mahmoud, for your input and comments. Initially, I was thought that the second manuscript is not related to ours. Now, I realized that the article is contextually related to our study under the umbrella of disaster (COVID-19 vs Wartime). I incorporated the article into my manuscript. (See line 94-98 in the manuscript). In addition to this, the whole manuscript was given to English experts in the University for proofreading, and amendments were made based on their feedback (see the revised manuscript with track changes)

Response to reviewer 1 comments and suggestions

In my first review of this article, I made comments on the need for post-hoc power analysis. This was included. Furthermore, I commended the authors of the important contribution of the article related to children and Covid-19 prevention behaviors and recommended authors to add relevant practical recommendations. What is unclear is the phrase "behavior change barograms" (line 488). Lastly, this revised submission still requires professional proofreading and editing to improve English expressions and grammar.

 Response: Thank you Dr. Seong-Yuen Toh for your valuable comments and suggestions. Sorry for the error. We said "behavior change barograms" just to say "behavior change programs", it was a typographical error. We made corrections accordingly (See line 461 in the manuscript). In addition to this proofreading of the whole manuscript was done by the research team and English language experts. Moreover, our study findings claimed that the COVID-19 preventive practice among students was quite low. Based on these findings, we recommended schools design and implement behavior change interventions to enhance their pupil’s COVID-19 prevention practice. We also suggest some important focus of intervention based on our finding including reduction of barriers through the provision of facemasks, hand sanitizers, and soaps for those who are in financial hardship and enhancing students' self-efficacy to improve their prevention practice (see the conclusion and recommendation section of the manuscript). 

Response to reviewer 2 comments and suggestions

I am pleased to see that that the authors have taken the previous review comments on board and have set out to deal with every one of them. There has clearly been a great deal of additional work undertaken to address each of the reviewers' concerns, which is duly noted and appreciated. This has made a significant impact on the quality of the paper. Thank you

Response: Thank you very much for your valuable contribution to the improvement of the manuscript.

---

## [Decision Letter · Decision Letter 2]

9 Jul 2021

PONE-D-21-05732R2

Application of Health Belief Model for the Assessment of COVID-19 Preventive Behavior and its Determinants among Students: A Structural Equation Modeling Analysis

PLOS ONE

Dear Dr. Shitu,

Thank you for submitting your manuscript to PLOS ONE. After careful consideration, we feel that it has merit but does not fully meet PLOS ONE’s publication criteria as it currently stands. Therefore, we invite you to submit a revised version of the manuscript that addresses the points raised during the review process.

We look forward to receiving your revised manuscript.

Kind regards,

Ali B. Mahmoud, Ph.D.

Academic Editor

PLOS ONE

Journal Requirements:

Reviewers' comments:

Reviewer's Responses to Questions

**Comments to the Author**

1. If the authors have adequately addressed your comments raised in a previous round of review and you feel that this manuscript is now acceptable for publication, you may indicate that here to bypass the “Comments to the Author” section, enter your conflict of interest statement in the “Confidential to Editor” section, and submit your "Accept" recommendation.

Reviewer #1: All comments have been addressed

Reviewer #3: All comments have been addressed

2. Is the manuscript technically sound, and do the data support the conclusions?

Reviewer #1: Yes

Reviewer #3: Yes

3. Has the statistical analysis been performed appropriately and rigorously? 

Reviewer #1: Yes

Reviewer #3: Yes

4. Have the authors made all data underlying the findings in their manuscript fully available?

Reviewer #1: Yes

Reviewer #3: Yes

5. Is the manuscript presented in an intelligible fashion and written in standard English?

Reviewer #1: Yes

Reviewer #3: Yes

6. Review Comments to the Author

Reviewer #1: Acceptable standard of English after the amendments and improvements.

Please accept this paper for publication.

Reviewer #3: This revision is an improved one. The paper needs detailed proof reading to reconcile English discrepancies. Please see my suggested modifications in the attached file.

7. PLOS authors have the option to publish the peer review history of their article (what does this mean?). If published, this will include your full peer review and any attached files.

Reviewer #1: No

Reviewer #3: No

---

## [Author Response · Author response to Decision Letter 2]

4 Aug 2021

Dear Ali B. Mahmoud, Ph.D., editor and reviewers:

Hereby, we resubmit the enclosed –revised- manuscript ID PONE-D-21-05732-[ EMID:15744fc9849b3e3d], which is entitled “Application of Health Belief Model for the Assessment of COVID-19 Preventive Behavior and its Determinants among Students: A Structural Equation Modeling Analysis” to your journal. We have read the comments carefully and we were able to implement all of them (see the manuscript with track changes). While reading the manuscript critically, we spotted English grammar errors, which we corrected. We hope that our revision will be felt like an improvement. Furthermore, we assessed our reference list for retracted papers and we didn’t cite any retracted article. Thank you!

Response to reviewers and editors

1) Suggestion to improve paper’s structure:

I suggest removing the first paragraph that introduces Covid-19. I find it to be too medically focused. There is no need to speculate on the origins of the virus (particularly since it is not established definitively) since it is irrelevant for the purpose of this paper. Alternatively, authors could replace medical wording with simple reference to Covid-19 pandemic. Everyone knows and understands what this is. 

I would suggest instead to incorporate some of the already established early research studies on negative effects of Covid-19 as applicable to either the same settings (such as educational environment), or the same demographics (such as kids). For example, the following paper summarizes the up-to-date research finding based on kids at-home eating habits during Covid:

“The Janus-faced effects of COVID-19 perceptions on family healthy eating behavior: Parent’s negative experience as a mediator and gender as a moderator” by Ali B. Mahmoud, Dieu Hack-Polay, Leonora Fuxman, Maria Nicoletti https://onlinelibrary.wiley.com/doi/10.1111/sjop.12742

Giving the reader the summary of findings about Covid related impacts within similar context would present a better introductory background to the current study. 

Response: Correction has been made accordingly (See line 67-82 on page 4 in the manuscript).

2) I suggest deleting the entire section “Study Period and Setting”. The only immediately relevant information in this section are the dates of the study, which authors can easily incorporate into the next section. 

Response: Correction has been made accordingly (See line 156-165 at page 5 in the revised manuscript with track changes).

This revision is an improved one. The paper needs detailed proof reading to reconcile English discrepancies. Please see my suggested modifications in the attached file.

Response: Thank you for taking time to review our work for the language details. Proofreading and language review have been made throughout the manuscript (See the revised manuscript with a track change).

3) Here are some suggestions to correct language throughout the manuscript

Line 73: remove end of sentence punctuation (See line 69-70 on page 4 in the manuscript).

Response: Correction has been made accordingly

Lines 75-78 (third paragraph): I don’t see the need to re-state mortality statistics as those do change rather quickly and are widely accessible online. This paper is about students in University setting and as such the focus of the introduction should be on the specifics of the environment under consideration. 

Line 88-89: grammar correction : “more than 94% of students were following the recommended preventive health behaviors in Iran 

Response: Correction has been made accordingly (See line 93-94 on page 5 in the manuscript).

Line 106: correct punctuation – remove “; ”

Response: Correction has been made accordingly (See line 112 on page 6 in the manuscript).

Line 116: grammar correction – “The HBM is one of the various health behavior models that has been…”

Response: Correction has been made accordingly (See line 122 on page 6 in the manuscript).

Lie 127: replace “study area” with academia.

Response: Correction has been made accordingly (See line 134 on page 7 in the manuscript).

Line 136: replace “administration” with administered.

Response: Correction has been made accordingly 

Line 147: replace “refers” with refer.

Response: Correction has been made accordingly (See line 145 on page 7 in the manuscript).

Line 151: and/or

Response: Correction has been made accordingly (See line 149 on page 7 in the manuscript).

Line 156: capitalize the title of the journal

Response: Correction has been made accordingly (See line 153 on page 7 in the manuscript).

Line 162: replace punctuation “;” with :

Response: Correction has been made accordingly (See line 159 on page 8 in the manuscript).

Line 163: coma needed ” … variable (0.43),” 

Response: Correction has been made accordingly (See line 160 on page 8 in the manuscript).

Line 167: “governmental schools, which resulted…”

Response: Correction has been made accordingly (See line 164 on page 8 in the manuscript).

Line 188: replace punctuation “;” with :

Response: Correction has been made accordingly (See line 183 on page 9 in the manuscript).

Line 190: insert “is resource efficient”

Response: Correction has been made accordingly (See line 185 on page 9 in the manuscript).

Line 235: delete coma - “practice concerning handwashing, …” 

Response: Correction has been made accordingly (See line 235 on page 11 in the manuscript).

Line 244: add coma – “cleaning, coding…”

Response: Correction has been made accordingly (See line 239 on page 11 in the manuscript).

Line 262: insert “ produce precise estimate with smaller…”

Response: Correction has been made accordingly (See line 257 on page 12 in the manuscript).

Line 276: add coma – “perceived benefit, perceived barrier…”

Response: Correction has been made accordingly (See line 271 on page 13 in the manuscript).

Lin 301: add reference for Declaration of Helsinki. 

Response: The reference has been added ((See line 296 on page 14 in the manuscript).

Line 312: replace “kept” with keep for grammatical consistency.

Response: Correction has been made accordingly (See line 307 on page 16 in the manuscript).

Line 324: correct - “…scores were higher than the neutral score…”

Response: Correction has been made accordingly (See line 319 on page 18 in the manuscript).

Line 337: add coma and lower case – “perceived severity (r = 0.2, p < 0.05), perceived benefit…”

Response: Correction has been made accordingly (See line 332 on page 18 in the manuscript).

All statistical techniques, new and old, should have reference. For example, line 336 and 365

Response: Correction has been made accordingly (See line 331 and 361 on page 18 and 361 in the manuscript, respectively).

Line 407-408: poor choice of wording “These results are results lower than the result reported by various previous studies…” Please rewrite this sentence 

Response: Correction has been made accordingly (See line 402-403 on page 24 in the manuscript).

Line 420: lower case “perceived barriers” 

Response: Correction has been made accordingly (See line 415 on page 25 in the manuscript).

Line 437: “…studies that were done using the health belief model…”

Response: Correction has been made accordingly (See line 432 on page 25 in the manuscript).

Line 440: use abbreviation HBM instead of health belief model.

Response: Correction has been made accordingly (See line 435 on page 25 in the manuscript).

Line 446: remove “is” or add our results “… the results are consistent…”

Response: Correction has been made accordingly (See line 441 on page 26 in the manuscript).

Line 449: replace punctuation “;” with:

Response: Correction has been made accordingly (See line 444 on page 26 in the manuscript).

Line 457: capitalize the name of the model to keep consistent “Health Belief Model”

Response: Correction has been made accordingly (See line 452 on page 26 in the manuscript).

---

## [Decision Letter · Decision Letter 3]

24 Jan 2022

Application of Health Belief Model for the Assessment of COVID-19 Preventive Behavior and its Determinants among Students: A Structural Equation Modeling Analysis

PONE-D-21-05732R3

Dear Dr. Shitu,

We’re pleased to inform you that your manuscript has been judged scientifically suitable for publication and will be formally accepted for publication once it meets all outstanding technical requirements. **Please, ensure the final version of your manuscript addresses the minor syntax edits suggested by Reviewer 3 and has undergone top-down proofreading.**

Kind regards,

Ali B. Mahmoud, Ph.D.

Academic Editor

PLOS ONE

Additional Editor Comments (optional):

Reviewers' comments:

Reviewer's Responses to Questions

**Comments to the Author**

1. If the authors have adequately addressed your comments raised in a previous round of review and you feel that this manuscript is now acceptable for publication, you may indicate that here to bypass the “Comments to the Author” section, enter your conflict of interest statement in the “Confidential to Editor” section, and submit your "Accept" recommendation.

Reviewer #3: All comments have been addressed

2. Is the manuscript technically sound, and do the data support the conclusions?

Reviewer #3: Yes

3. Has the statistical analysis been performed appropriately and rigorously? 

Reviewer #3: Yes

4. Have the authors made all data underlying the findings in their manuscript fully available?

Reviewer #3: Yes

5. Is the manuscript presented in an intelligible fashion and written in standard English?

Reviewer #3: Yes

6. Review Comments to the Author

Reviewer #3: please make corrections in lines:

73: "Schools"– sentence case

147: what is “after home checkups”?

201: extra parenthesis

7. PLOS authors have the option to publish the peer review history of their article (what does this mean?). If published, this will include your full peer review and any attached files.

Reviewer #3: No

---

## [Editor Report · Acceptance letter]

11 Mar 2022

PONE-D-21-05732R3 

Application of Health Belief Model for the Assessment of COVID-19 Preventive Behavior and its Determinants among Students: A Structural Equation Modeling Analysis 

Dear Dr. Shitu:

I'm pleased to inform you that your manuscript has been deemed suitable for publication in PLOS ONE. Congratulations! Your manuscript is now with our production department. 

Kind regards, 

on behalf of

Dr. Ali B. Mahmoud 

Academic Editor

PLOS ONE